# Clinical and Genetic Characteristics of Pediatric Patients with Hypophosphatasia in the Russian Population

**DOI:** 10.3390/ijms232112976

**Published:** 2022-10-26

**Authors:** Oleg S. Glotov, Kirill V. Savostyanov, Tatyana S. Nagornova, Alexandr N. Chernov, Mikhail A. Fedyakov, Aleksandra N. Raspopova, Konstantin N. Krasnoukhov, Lavrentii G. Danilov, Nadegda V. Moiseeva, Roman S. Kalinin, Victoria V. Tsai, Yuri A. Eismont, Victoria Y. Voinova, Alisa V. Vitebskaya, Elena Y. Gurkina, Ludmila M. Kuzenkova, Irina B. Sosnina, Alexander A. Pushkov, Ilya S. Zhanin, Ekaterina Y. Zakharova

**Affiliations:** 1Department of Genomic Medicine, D. O. Ott Research Institute of Obstetrics, Gynaecology and Reproductology, 199034 Saint-Petersburg, Russia; 2Department of Experimental Medical Virology, Molecular Genetics and Biobanking of Pediatric Research and Clinical Center for Infectious Diseases, 197022 Saint-Petersburg, Russia; 3National Medical Research Center of Children Health, 119991 Moscow, Russia; 4Research Center for Medical Genetics, 115478 Moscow, Russia; 5Bioenergetics Department of Life Sciences, The National Institute for Biotechnology in the Negev, Ben-Gurion University of the Negev, Beer-Sheva 84105, Israel; 6Department of General Pathology and Pathological Physiology, Institute of Experimental Medicine, 197376 Saint-Petersburg, Russia; 7CerbaLab Ltd., 199106 Saint-Petersburg, Russia; 8City Hospital No. 40, St.-Petersburg, 9 Borisova Str., Sestrorezk, 197706 Saint-Petersburg, Russia; 9Department of Genetics and Biotechnology, Saint-Petersburg State University, 199034 Saint-Petersburg, Russia; 10Veltischev Research and Clinical Institute for Pediatrics, Pirogov Russian National Research Medical University, 117997 Moscow, Russia; 11Department of Children’s Diseases of N.F. Filatov Clinical Institute for Children’s Health, I.M. Sechenov First Moscow State Medical University, 119991 Moscow, Russia; 12Children’s Rehabilitation Clinic Almazov National Medical Research Centre, 197341 Saint-Petersburg, Russia; 13Saint-Petersburg State Budgetary Healthcare Institution “Consulting and Diagnostic Center for Children”, 192289 Saint-Petersburg, Russia

**Keywords:** hypophosphatasia, clinical forms of hypophosphatasia, tissue-specific alkaline phosphatase, *ALPL* gene, SNP, residual activity of *ALPL* mutant alleles

## Abstract

(1) Hypophosphatasia (HPP) is a rare inherited disease caused by mutations (pathogenic variants) in the *ALPL* gene which encodes tissue-nonspecific alkaline phosphatase (TNSALP). HPP is characterized by impaired bone mineral metabolism due to the low enzymatic activity of TNSALP. Knowledge about the structure of the gene and the features and functions of various *ALPL* gene variants, taking into account population specificity, gives an understanding of the hereditary nature of the disease, and contributes to the diagnosis, prevention, and treatment of the disease. The purpose of the study was to describe the spectrum and analyze the functional features of the *ALPL* gene variants, considering various HPP subtypes and clinical symptoms in Russian children. (2) From 2014–2021, the study included the blood samples obtained from 1612 patients with reduced alkaline phosphatase activity. The patients underwent an examination with an assessment of their clinical symptoms and biochemical levels of TNSALP. DNA was isolated from dried blood spots (DBSs) or blood from the patients to search for mutations in the exons of the *ALPL* gene using Sanger sequencing. The PCR products were sequenced using a reagent BigDye Terminator 3.1 kit (Applied Biosystems). Statistical analysis was performed using the GraphPad Prism 8.01 software. (3) The most common clinical symptoms in Russian patients with HPP and two of its variants (*n* = 22) were bone disorders (75%), hypomyotonia (50%), and respiratory failure (50%). The heterozygous carriage of the causal variants of the *ALPL* gene was detected in 225 patients. A total of 2 variants were found in 27 patients. In this group (*n* = 27), we identified 28 unique variants of the *ALPL* gene, of which 75.0% were missense, 17.9% were frameshift, 3.6% were splicing variants, and 3.6% were duplications. A total of 39.3% (11/28) of the variants were pathogenic, with two variants being probably pathogenic, and 15 variants had unknown clinical significance (VUS). Among the VUS group, 28.6% of the variants (7/28) were discovered by us for the first time. The most common variants were c.571G > A (p.Glu191Lys) and c.1171del (Arg391Valfs*12), with frequencies of 48.2% (13/28) and 11% (3/28), respectively. It was found that the frequency of nonsense variants of the *ALPL* gene was higher (*p* < 0.0001) in patients with the perinatal form compared to the infantile and childhood forms of HPP. Additionally, the number of homozygotes in patients with the perinatal form exceeded (*p* < 0.01) the frequencies of these genotypes in children with infantile and childhood forms of HPP. On the contrary, the frequencies of the compound-heterozygous and heterozygous genotypes were higher (*p* < 0.01) in patients with infantile childhood HPP than in perinatal HPP. In the perinatal form, residual TNSALP activity was lower (*p* < 0.0005) in comparison to the infantile and childhood (*p* < 0.05) forms of HPP. At the same time, patients with the heterozygous and compound-heterozygous genotypes (mainly missense variants) of the *ALPL* gene had greater residual activity (of the TNSALP protein) regarding those homozygous patients who were carriers of the nonsense variants (deletions and duplications) of the *ALPL* gene. Residual TNSALP activity was lower (*p* < 0.0001) in patients with pathogenic variants encoding the amino acids from the active site and the calcium and crown domains in comparison with the nonspecific region of the protein.

## 1. Introduction

Hypophosphatasia (HPP, OMIM: 146300, 241500, 241510) is a rare inherited mineral metabolism disorder caused by low enzymatic activity in the tissue nonspecific alkaline phosphatase (TNSALP) as a result of mutations (pathogenic variants) in the *ALPL* gene that is localized on chromosome 1p36.12 [1]. The prevalence of HPP is variously estimated at 1/100,000–1/900,000 liveborn [2]. The incidence of pathogenic variants in the *ALPL* gene in the Russian population is 4.2% [3]. HPP was first mentioned in 1948, and the first pathogenic variant of c. 711A > G (Ala162Thr) in the *ALPL* gene was detected in 1988 [4,5].

A high level of TNSALP expression is observed in the liver, kidneys, and bones [6]. TNSALP is involved in the dephosphorylation of pyridoxal-5-phosphate (PLP) and inorganic pyrophosphate (PPi) in these organs and tissues. A decrease in its serum activity leads to the accumulation of PPi mineralization inhibitors in tissues, which causes bone and tooth defects [7]. It has been demonstrated that a decrease in TNSALP activity and the concentration of its substrates correlate with the severity of the clinical symptoms of HPP, defining a heterogeneous phenotype of the disease [8]. However, the activity of TNSALP can decrease under the influence of drugs, the lack of zinc, magnesium (in celiac disease), achondroplasia, and anemia and lead to an increase in liver diseases and osteogenesis imperfecta [9].

HPP is usually classified into six clinical forms: perinatal, prenatal benign, infantile, childhood, odonto-HPP, and adult [10]. The most severe perinatal form of HPP manifests in utero or can be obviously diagnosed at birth based on profound skeletal hypomineralization, caput membraneceum, shorted and deformed limbs, and pyridoxine-dependent seizures. These patients have a very high risk of death in the first days of life due to the hypomineralization of the ribs, pulmonary hypoplasia, and asphyxia [11]. The infantile type is characterized by respiratory complications, total demineralization, skeletal deformities, and rickets, with symptoms present in the first six months of life [12]. The childhood form of HPP appears between 6 months and 18 years of age, with mineralization defects, bone deformities, premature loss of deciduous teeth, delayed walking, short stature, brain damage, delayed motor development and skill formation, and the progression of convulsions [13]. Odonto-HPP is typical in early isolated damage to the dentition, with the premature loss of deciduous teeth, impaired teeth growth and structure, and periodontal disease. Adult HPP expresses skeletal hypomineralization leading to fractures in the metatarsal bones and femur, with premature tooth loss, muscle and joint pain, arthropathy, pseudogout and enthesopathy, and neurological symptoms [12].

The clinical symptoms and severity of HPP, however, can vary considerably in different patients, even within the same form of the disease, which is associated with the heterogeneous pathogenic variants in the *ALPL* gene [14,15]. For this reason, the diagnosis of HPP involves measuring TNSALP activity levels in the serum according to age and gender and searching for mutations in the *ALPL* gene (NM_000478.6, 12 exons, and a length of about 70,000 bp) [16].

By February 2022, more than 500 pathogenic variants of the *ALPL* gene were registered in the Global Variome Shared International Database [17]. Among all *ALPL* genetic variants, missense mutations account for 71.2%, deletions 11.0%, splice mutations 4.9%, nonsense mutations 4.6%, insertions 3.4%, large deletions/duplications 2.9%, insertions/deletions 1.5%, and regulatory mutations 0.2% [18].

The missense variants of the *ALPL* gene can affect the level of expression, modification, folding, membrane transport, and dimerization of TNSALP, which is phenotypically indicated by different levels of its residual enzymatic activity [19]. At the same time, it has been established that the severity of HPP depends on the inheritance type. In mild forms, there is a functional loss of a single copy of the *ALPL* gene, which is reflected in the autosomal-dominant inheritance of the disease, with varying expression and incomplete penetrance. The perinatal and infantile forms are characterized by recessive inheritance [10]. Meanwhile, the pathogenic variants of the *ALPL* gene can exhibit a dominant-negative effect (DNE) due to the mutant monomer inhibiting the interaction between the wild-type monomers or due to the sequestering of the wild-type protein by the mutant monomer, preventing its transport to the membrane, thereby explaining the presence of the dominant forms of the disease [20]. Some results demonstrate that the in vitro residual enzymatic activity of the mutant protein and/or DNE value correlates with the clinical symptoms of HPP [21]. Study objective: to describe the spectrum and analyze the functional features of the variants of the *ALPL* gene with regard to different HPP subtypes and clinical symptoms in Russian children.

## 2. Results

### 2.1. Clinical Features and Forms of Hypophosphatasia

The clinical symptoms of HPP in children with variant(s) in the *ALPL* gene in the Russian population were most fully described in only 56 of the patients, and clinical information was available for 22 compound heterozygotes carriers of the causal variants in these patients (Table 1, Figure 1).

Among children with perinatal HPP (*n* = 4), the most frequent symptoms were bone deformity (75%), craniosynostosis (50%), pectus excavatum (50%), bone hypomineralization (50%), limb shortening (50%)), skull bone deformity (75%), skull hypomineralization (75%), hypotension (50%), and respiratory failure (50%) (Figure 2).

For children with infantile HPP (*n* = 8), the most common symptoms were skeletal deformities (87.5%), tooth loss (75.0%), neuropsychomotor developmental delay (75.0%), and malnutrition (50.0%) (Figure 3).

In patients with childhood HPP (*n* = 10), the most frequent symptoms were tooth loss (100.0%), hypomyotonia (60.0%), skeletal deformity (40.0%), gait disorder (40.0%), muscle and joint pain (40.0%) neuropsychomotor developmental delay (40.0%), and fatigue (40.0%) (Figure 4).

### 2.2. Variant Frequencies in the ALPL Gene at HPP

Compound heterozygotes were found in 27 of the patients with HPP. A total of 28 different genetic variants were found in this group of patients, including some previously undescribed in other populations (Table 2). The variants were classified according to the American College of Medical Genetics and Genomics (ACMG) guidelines (Richards et al., 2015).

In bold are the variants that are not represented in the Global Variome shared LOVD *ALPL* database as of 1 August 2022 (VUS: a variant of uncertain (or unknown) significance).

The most commonly found variants among those patients with two variants were c.571G > A(p.Glu191Lys) and c.1171del (Arg391Valfs*12) with frequencies of 48.2% (13/28) and 11% (3/28), respectively. These variants are localized in exons 6 and 10, correspondingly. According to the ACMG classification [15], for this group of patients, 71.4% (20/27) of the genetic variants were identified as pathogenic, and 8 variants (28.6%) were of an uncertain (or unknown) significance (VUS). Among these variants, 25.0% (7/28: 4 pathogenic and 3 VUS) were absent from the LOVD and dbSNP databases and were initially identified in our study (Table 2). 

The heterozygous carriage of the variants was detected in 225 of the total 1612 patients with decreased alkaline phosphatase activity. A total of 66 unique genetic variants were identified (Table 3).

In bold are variants that are not represented in the Global Variome shared LOVD *ALPL* database as of 1 August 2022 (VUS a variant of uncertain (or unknown) significance). All genetic variants are presented in Appendix A in the Appendix A.

The most common variants in heterozygous patients were also c.571G > A (p.Glu191Lys), 1171del (Arg391Valfs*12), and c.455G > A (Arg152His) with frequencies of 12.7, 3.1, and 3.3% (57/450, 14/450, 15/450), respectively. These variants were localized in exons 5, 6, and 10. According to the ACMG [17], for the heterozygous patients, 60.6% (40 of 66) of the genetic variants were pathogenic, one variant (1.5%) was likely pathogenic or pathogenic, and 25 variants (37.9%) were VUS. The 18 other variants (27.3%, 18/66) were not previously described in the LOVD and dbSNP databases and literature and were first identified in this research.

### 2.3. Analysis of the Variants Depending on Type and Localization in the ALPL Gene

Compound hererozygotes in the *ALPL* gene were detected in 27 of the individuals. The majority of them were missense (75.0%), with 17.9% frameshift and in frame, 3.6% splice variants, and 3.6% duplications (Figure 5).

Interestingly, 50.0% of the variants (14/28) in the patients with two variants were localized in the 5th (6 variants), 10th (4 variants), and 12th (4 variants) exons of the *ALPL* gene (Figure 6).

The distribution of amino acid substitutions in the TNSALP protein domains in patients with the two variants is illustrated in Figure 7.

Interestingly, the two variants (Ser368del and Glu455Asp) were localized in the active center of the enzyme. The Glu291Gln mutation was located in the calcium-binding domain, and mutations Tyr388Cys, Arg391ValfsTer12, Arg455His, and GGYTPRG386_392ATGGVST) were located in the corona domain. These mutations may also lead to decreased enzyme activity. Another mutation, Asp294Ala, located in the binding center of this domain leads to pathological changes [22]. There were also seven mutations, Val50AlafsTer20, Gly61Glu, Arg71Cys, Gln76His, Thr85Pro, Val459Met, and Val483Met, located in the homodimer interaction site. These mutations may cause the disruption of homodimer formation and, as a consequence, the disruption of protein incorporation into the cell membrane.

### 2.4. Analysis of ALPL Gene Variants Depending on the Form of HPP and Clinical Symptoms in Patients

We analyzed the genotypes of the patients with two pathogenic and VUS forms (*n* = 22) separately, according to their most severe clinical symptoms and the form of HPP (Table 4).

A total of three out of four patients with the perinatal form, according to the data in Table 4, were homozygotes (two in the missense and one in the nonsense variants) and had bone disorders and skull hypomineralization; one child had hypotonia and another had respiratory disorders. Of the eight patients with the infantile form, six were compound heterozygotes and two were heterozygotes of the missense and nonsense variants. This group of patients was predominated by skeletal deformities (87.5%), loss of teeth (75.0%), and neuropsychomotor developmental delay (75.0%). Of the ten patients with the pediatric form, seven were compound heterozygotes, two were splice heterozygotes, and one was nonsense heterozygote. Tooth loss (100.0%) and hypotonia (60.0%) prevailed in this group of patients. It is noteworthy that the homozygous patients, regarding missense variants or compound heterozygotes for nonsense variants, had severe clinical symptoms.

Further, we compared the frequencies of missense and nonsense variants depending on the form of HPP (Table 3, Figure 8).

As follows from the data presented in Figure 4, there is not a statistically significant (*p* > 0.05) predominance of the nonsense mutations of the *ALPL* gene in patients with the perinatal form compared with the infantile and childhood forms of HPP.

We also compared the frequencies of the homozygous and heterozygous variants of the *ALPL* gene in different forms of HPP (Table 3, Figure 9).

The number of homozygotes with the perinatal form was significantly (*p* < 0.01) higher than the number of these genotypes in patients with the infantile and childhood forms of HPP. On the contrary, the frequency of the compound heterozygotes and heterozygotes was higher (*p* < 0.01) in patients with the infantile and childhood forms when compared to the perinatal form of HPP.

### 2.5. Analysis of TNSALP Activity Depending on the Form, Type of Variants, and the Genotype of the Patients

We analyzed the TNSALP activity, depending on the form of HPP (Table 3, Figure 10 and Figure 11).

Based on the works of [21], we compared the theoretical residual activity of the variants in our group of patients according to its TNSALP domain localization (Figure 11).

The results presented in Figure 10 and Figure 11 establish that residual TNSALP activity in the perinatal form was statistically significantly (*p* < 0.0005) lower than in the infantile and pediatric (*p* < 0.05) forms of HPP.

However, the theoretically calculated (the real data for these groups of patients are partially available), heterozygotes and compound heterozygotes, with variants in the *ALPL* gene (Table 4, Figure 12A), have higher residual protein activity compared to the homozygotes. We did not detect differences between the real age-related changes and the theoretically-calculated activity of the TNSALP for different genotypes (Figure 12B).

### 2.6. Analysis of the Activity of the ALPL Gene Variants Depending on Their Domain Localization and Origin

We compared the activity of the *ALPL* gene variants in our group of patients depending on their domain localization (Figure 13).

It was found that the activity of TNSALP for those variants encoding the amino acids in the “active center” and corona domain was statistically significantly (*p* < 0.05) lower than the activity in the variants at the homodimeric interface (nonspecific sites of the protein). This implies that these sites in the protein structure are crucial for the preservation of enzyme activity.

## 3. Discussion

In our study of Russian children with HPP, the most frequent symptoms were tooth loss (72.7%), skeletal deformities (63.6%), hypotonia (50.0%), and neuropsychomotor developmental delay (50%) (Figure 1). Whereas, in the European and American childhood populations (*n* = 121), the main symptoms of HPP were premature tooth loss (48.2%), bone disorders (44.2%), craniosynostosis (26.7%), and pain syndrome (19.3%) [24]. Among the four Russian children with perinatal HPP, the most frequent symptoms were bone deformities (75%), skull bone deformity (75%), skull hypomineralization (75%), hypotonia (50%), craniosynostosis (50%), pectus excavatum (50%), bone hypomineralization (50%), limb shortening (50%), and respiratory failure (50%) (Figure 2). Whereas, for Canadian and American children with perinatal HPP (*n* = 30), the most frequent symptoms were fractures (40.0%), skull abnormalities (40.0%), and general motor difficulties (23.3%) [25]. Notably, there are common clinical features for different populations, as well as specific ones. The most common symptoms in the eight Russian children with infantile HPP were skeletal deformities (87.5%), loss of teeth (75.0%), neuropsychomotor developmental delay (75.0%), and nutritional disorders (50.0%) (Figure 3). In contrast, in a group of Canadian and American children with infantile HPP (*n* = 102), the leading symptoms were loss of teeth (61.8%), cranial malformations (43.1%), and motor and mobility disorders (40.2%) [25]. Among the 10 Russian patients with childhood HPP, the most frequent symptoms were loss of teeth (100.0%), hypotonia (60.0%), skeletal deformity (40.0%), gait disorders (40.0%), muscle and joint pain (40.0%) motor retardation (40.0%), and fatigue (40.0%) (Figure 4). In Canadian and American children with childhood HPP (*n* = 87), the main symptoms were loss of teeth (70.1%), fractures (34.5%), and pain syndrome (43.7%) [25]. Note that those patients with more severe symptoms (skeletal deformities, loss of teeth, etc.) in the Russian, American, and European populations have a homozygous genotype or have nonsense variants (deletions or splice variants) in the genotype. In general, the low frequency of HPP detection in the Russian population is due to the fact that the main criterion for selecting patients was not the presence of specific clinical symptoms but a recorded low level of alkaline phosphatase, which could not be associated with HPP. The difference in the frequency of the clinical symptoms in Russian patients compared with foreign data is due to (1) a small sample of our patients with a confirmed diagnosis, for whom the detailed clinical data were available, (2) the retrospective analysis of clinical data from the medical record, and (3) a lack of uniform criteria for assessing some symptoms (for example, if a child did not complain of pain by their own accord, they could not be asked about it).

Additionally, we identified a total of 79 unique variants of the *ALPL* gene, characterized by residual enzymatic activity, from 252 Russian patients with HPP. Among them, patients with two pathogenic variants had 28 different variants. Of these, 75% of the patients had missense variants, 17.9% had frameshift deletions, 3.6% had duplications, and 3.6% had splicing variants. American and French researchers obtained similar frequencies for their patients. They found a total of 155 variants of the *ALPL* gene in 345 patients with HPP. Missense variants accounted for 146 (94.2%) patients, nonsense variants (variants with a broken chain) for five (3.2%), variants with a reading frame shift for two (1.3%), and deletions for two (1.3%) patients [23]. It should be noted that our group of patients was less characterized by missense variants, while the reading frameshift deletions were more representative. This is probably associated with the fact that our group included patients both with two variants and/or VUS variants and with a more severe form of HPP.

The c.571G > A(p.Glu191Lys) variant in the 6th exon in the *ALPL* gene was the most frequent among the compound-heterozygote and heterozygotes in the Russian population at 28.89% (73/252). According to the study by del Angel G. et al., the frequency of this c.571G > A/p.Glu191Lys variant is more than 10^−3^ [23]. The c.571G > A/p.Glu191Lys variant is also the most common (with a frequency of up to 55%) in patients with HPP of a European origin [26], and the initial frequency of this variant was 0.08 in infants, children, and patients with odonto-HPP [6]. The c.571G > A/p.Glu191Lys variant has been demonstrated to reduce TNSALP enzymatic activity by up to 68% (207.1 ± 39.3 U/μg) when compared with wild-type enzyme activity (302.5 ± 102.2 U/μg) [26].

The other variants, such as c.1001G > A (p.Gly334Asp), both in our study and in the study by del Angel G. et al. [23], were detected only in patients with HPP at a carrier frequency of less than 1%, which corresponds to the frequency of this variant in Manitoba Mennonites [27]. Finally, it should be noted that some variants can be detected with a higher frequency in a particular population. For example, the frequency of the c.787T > C/p.Tyr263His variant in the Japanese population is 0.31–0.55, and in the gnomAD database, 0.1833 [28,29] of the p.Asp378Val and p.Leu520ArgfsX86 variants are most common in the USA [6] and Japan [30], respectively. The majority of the mutations in the *ALPL* gene were localized in the 5th, 7th, 10th, and 3rd exons [18].

In the study by del Angel G. et al. (2020), the pathogenic variants (reading frame shift, deletions, and splicing variants) exhibited close to zero residual enzymatic activity. An interesting fact is that, whereas low residual enzymatic activity can serve as a predictor of pathogenicity, the converse statement is incorrect. High residual activity values for the variants are not necessarily a predictor of their benignity. In calculating the activity within patients with HPP, it was found that extremely low in vitro values correlate with perinatal lethal or infantile HPP subtypes (Figure 7). At the same time, there was an overlap in activity between patients with HPP and healthy donors (≥50%) [21]. Consequently, the prognostic value of low TNSALP enzymatic activity is limited by extremely low values in the most severe HPP subtypes. Data obtained by gnomAD (version 2.1.1) [6] show that 844 out of 923 genetic variants are rare, with allele frequencies less than <10^−4^ in the population, independent of the residual activity level, which is compatible with the rare nature of HPP, especially in its perinatal and infantile subtypes [15,31]. Meanwhile, variants with frequencies of more than 10^−3^ have higher residual TNSALP activity [21]. This is related to evolutionary pressure, which results in variants with low activity occurring at low population frequencies. This conclusion is supported by the comparison of variant residual activity with clinical significance [32]. In addition, high values of residual enzymatic activity (>50% of WT values) can also be associated with the clinical subtype of HPP. Notably, pathogenic variants in the *ALPL* gene can have a dominant-negative effect (DNE) due to the inhibition of the interaction with the wild-type monomers by mutant monomers or as a result of the sequestration of the wild-type protein by the mutant monomer, preventing its transport to the membrane [33]. In our study, the residual enzymatic activity of TNSALP was most decreased for those variants associated with amino acid substitutions in the active center, corona domain, and the homodimeric interface.

We have determined residual enzymatic activity in 22 patients, including 14 patients (66.7%) with residual TNSALP activity of less than ≤25.0%, six (28.6%) with activity between 25–50%, and one patient with activity above 50%. In a study by del Angel G. et al., 58% (90/155) of the variants had a residual activity of less than ≤25%, 34% (52/155) of the variants exhibited an activity of more than ≥50%, and 8% (13/155) had activity between these levels [21].

Another study noted that pathogenic variants are localized at the calcium-binding site [34]. In our study, curiously, the residual activity of TNSALP at the homodimer interaction site covered a wide range. Consequently, the clinical significance of the variants in this domain may depend on the positional interactions and the specific amino acid changes involved. This conclusion is confirmed by comparing the residual activity of a variant to its clinical significance [32]. At the same time, high values of residual enzymatic activity (>50% of WT values) may also be associated with the clinical subtype of HPP. In addition, it is suggested that some alleles in patients with severe types of HPP may not reduce enzymatic activity but impair its transport and the localization of the TNSALP dimer in the extracellular domain or by impairing the creation of aggregates that are improperly cleaved in the proteasome [19,35].

## 4. Materials and Methods

### 4.1. Patients with Hypophosphatasia

Genetic testing was performed on 1612 patients, with decreased activity of alkaline phosphatase in the blood plasma from different cities and regions of the Russian Federation (Moscow, Saint Petersburg, Yekaterinburg, Surgut, Khabarovsk, Cheboksary, Krasnodar, etc.) from 2014–2021. The inclusion criteria for the patients were a detected low level of alkaline phosphatase, clinical symptoms, radiographic changes, and genetic variants in the *ALPL* gene. Heterozygous carriage of pathogenic and likely pathogenic variants in the *ALPL* gene was detected in 225 patients. Compound heterozygotes and homozygotes were found in 27 patients. The sex ratio of all subjects was approximately 3:1 (males to females). The average age of onset was 7 months (from 2 h after birth to 14 years old). The mean age of diagnosis was 2.43 years (2 h after birth to 19 years old).

### 4.2. Biochemical Studies

Patients included in this study were tested for screening diagnostic purposes. Fasting blood samples were collected from all patients to obtain serum. TNSALP activity in the serum was measured according to reference values, depending on age and sex using kinetic and colorimetric methods and an alkaline phosphatase kit (BioSystems S.A. biochemical analyzer) (Costa Brava, Barcelona Spain) [36]. The TNSALP concentration in the serum samples was calculated using the following formula:
$$\Delta A\times \frac{\mathrm{Vtx}\times 106}{\varepsilon \mathrm{xlxVs}}$$
where ∆A/min is the average absorption difference per minute, ε is the molar absorption coefficient of 4-nitrophenol at 405 nm, which is 18450, the optical path (l) is 1 cm, the total reaction volume (Vt) is 1.02, the sample volume (Vs) is 0.02, and 1 U/L is 0.0166 µkat/L. The following factors were used to calculate enzyme activity.
ΔA × 2764 = U/L or × 46.08 = µkat//L

TNSALP values in patients with two variants in the *ALPL* gene in different clinical forms of HPP are presented in Table 1.

### 4.3. Molecular Genetic Analysis of a ALPL Gene

The search for mutations in the exons of the *ALPL* gene was performed using Sanger sequencing. DNA was isolated from dry blood spots or whole blood by phenol-chloroform extraction. Special primers were developed for each exon in the *ALPL* gene. PCR conditions were selected individually for each exon. The synthesized PCR products were purified with 5 m NH4Ac and 96% ethanol, and then 70% ethanol, dried at 60 °C and dissolved in 10 µL of deionized water. Further, the purified PCR products were sequenced using the reagent BigDye Terminator 3.1 kit (Applied Biosystems, Waltham, MA, USA). Then, capillary electrophoresis was performed in a genetic analyzer 3500xl (Applied Biosystems, Waltham, MA, USA). The obtained exon sequences were analyzed using the Sequence Scanner (version 2.0, Applied Biosystems, Waltham, MA, USA) software. The presence of the mutation was confirmed by visual inspection of the sequencing chromatograms.

### 4.4. Predicting Theoretical Activity of Patient Genotypes

Predicting theoretical TNSALP residual activity in Russian HPP patients (*n* = 252) depended on genotype (homozygotes, heterozygotes, and compound heterozygotes) and was calculated as a fraction of WT activity for each patient. The TNSALP residual activity for the homozygous patients, which had a variant with complete loss of function (nonsense, splice, or frameshift), was assumed to be null. The TNSALP residual activity for the heterozygotes was detected as 50% from the activity of the homozygotes. In patients with a single LoF variant, the TNSALP residual activity was 0.5, with the suggestion that the LoF allele would not express any protein. The TNSALP residual activity for the compound heterozygotes was calculated as the average activity of both alleles. If an allele had a dominant negative effect (DNE), the TNSALP activity was determined as “activity divided by four”, or leaving only 25% from total TNSALP activity, with two copies of allele one [21].

### 4.5. Statistical Analysis

The statistical data were analyzed using GraphPad Prism 8.01 software. The Kolmogorov–Smirnov test was used to determine the type of data distribution. ANOVA, with posthoc (usually Tukey’s honest significant difference, HSD) used to find the differences between several normally-distributed samples. When pairwise comparing data with a distribution other than normal, the Mann–Whitney U test was used [37].

### 4.6. Bioinformatic Analysis

Variant frequencies in the population were analyzed using the gnomad v2.1.1 database, as well as reference datasets, such as 1000 Genome [38], ExAC [39], and the proprietary Russian exome database [40,41,42].

## 5. Conclusions

From 2014–2021, we performed genetic testing on 1612 patients from various regions of the Russian Federation, with decreased alkaline phosphatase activity in their blood plasma. Among them, we identified 75 variants of the *ALPL* gene characterized by residual enzymatic activity in 252 Russian patients with HPP. A total of 75% of these patients had missense variants, and the others had nonsense variants. Among them, 71.4% (20/28) of the genetic variants were identified as being pathogenic, and eight variants (28.6%) were of uncertain (or unknown) significance (VUS). Among these variants, 25.0% (7/28: four pathogenic and three VUS) were absent from the database and were initially identified by us.

The high frequency of hypotension in children from the Russian population is notable in the clinical picture, whereas the other clinical symptoms were similar. Among the heterozygous patients, 60.6% (40 of 66) of the genetic variants were pathogenic, one variant (1.5%) was likely pathogenic or pathogenic, and 25 variants (37.9%) were VUS. A total of 18 variants (27.3%: 18/66) were not previously described in the LOVD and dbSNP databases and the literature and were first identified in this research. The frequencies of the nonsense variants and low TNSALP enzymatic activity were higher in patients with the perinatal form of HPP compared to the infantile and childhood forms. Residual activity was detected as a function of the HPP subtype, variant type, and protein domain localization. TNSALP enzymatic activity cannot be used to assess the severity of HPP, but in combination with clinical symptoms, it will contribute to a better understanding of the phenotypic spectrum of HPP.

## Figures and Tables

**Figure 1 ijms-23-12976-f001:**
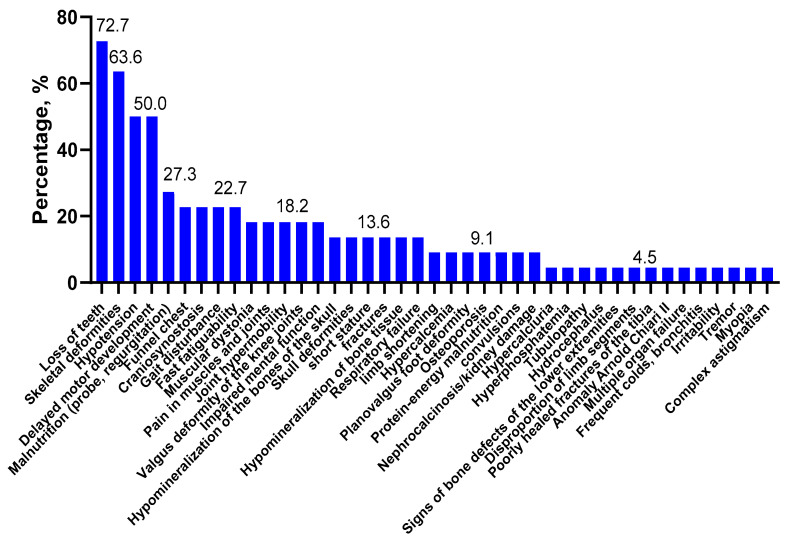
Clinical symptoms of HPP in the *ALPL* gene for patients with two pathogenic variants and within the Russian population (*n* = 22). The numbers above the columns indicate the percentage of the manifestation of the clinical symptom.

**Figure 2 ijms-23-12976-f002:**
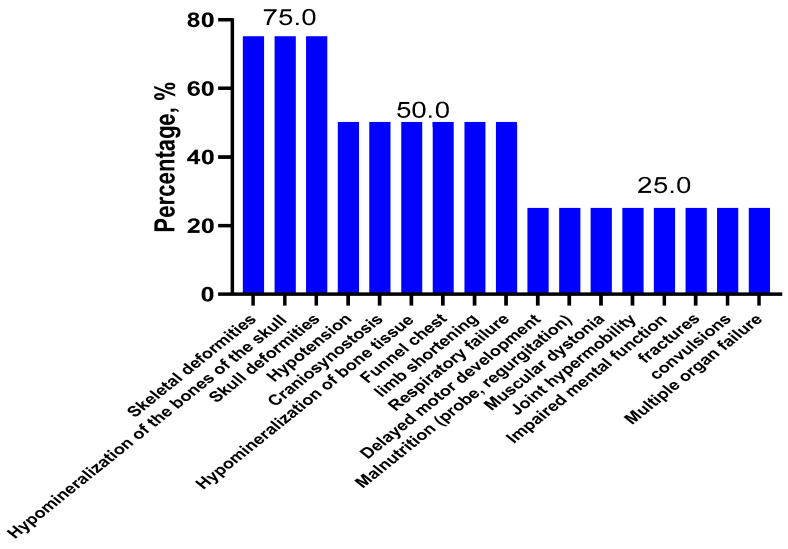
Clinical symptoms of HPP in children with perinatal HPP (*n* = 4).

**Figure 3 ijms-23-12976-f003:**
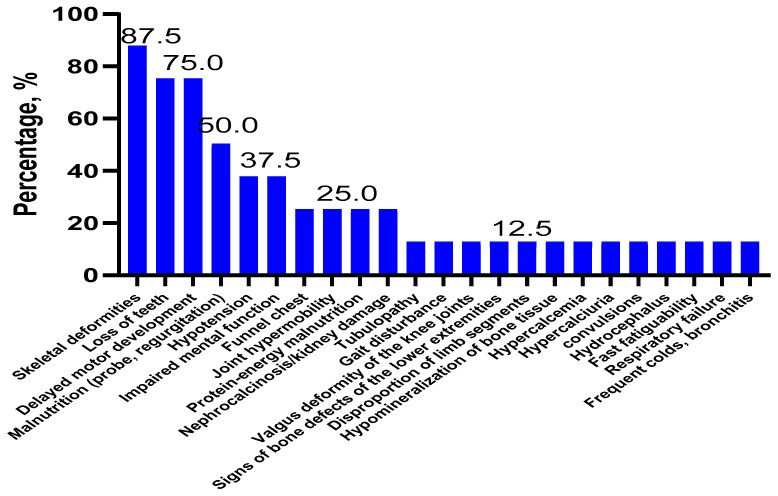
Clinical symptoms of HPP in children with infantile HPP (*n* = 8).

**Figure 4 ijms-23-12976-f004:**
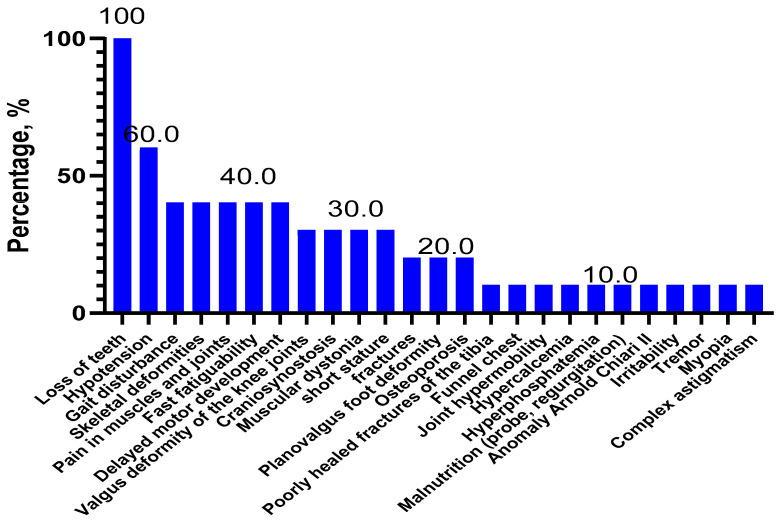
Clinical symptoms of HPP in patients with childhood HPP (*n* = 10).

**Figure 5 ijms-23-12976-f005:**
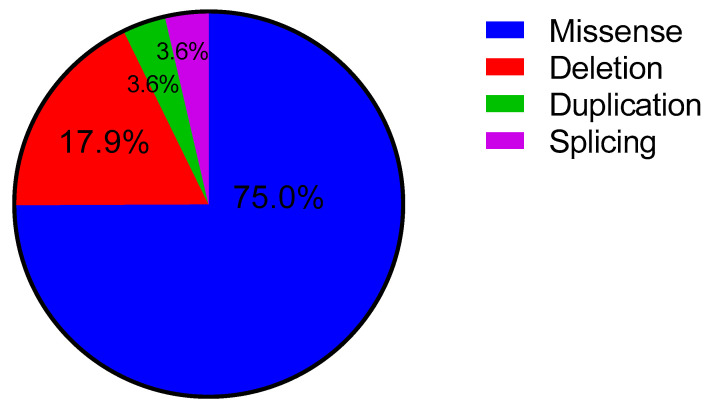
Distribution of variants depending on the type in the *ALPL* gene.

**Figure 6 ijms-23-12976-f006:**
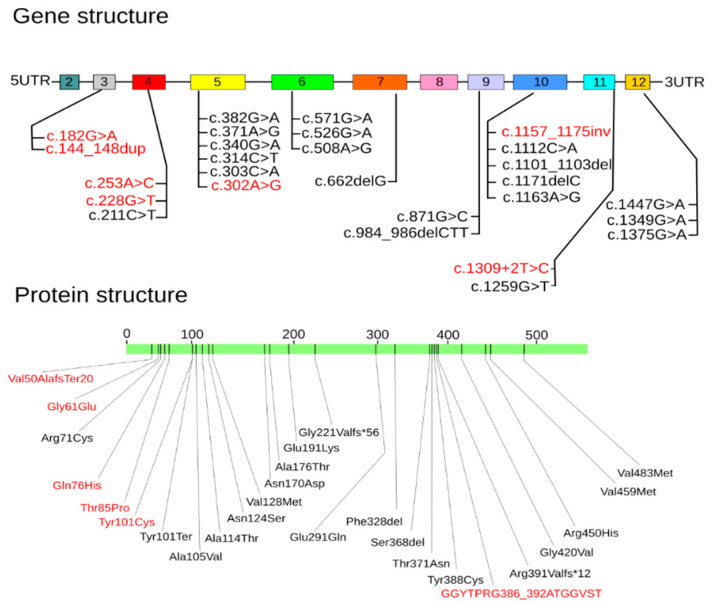
Distribution of genetic variants in the *ALPL* gene and amino acid substitutions in the TNSALP protein in Russian patients (compound heterozygotes and homozygotes only). Variants are highlighted in red, identified for the first time in the Russian population.

**Figure 7 ijms-23-12976-f007:**
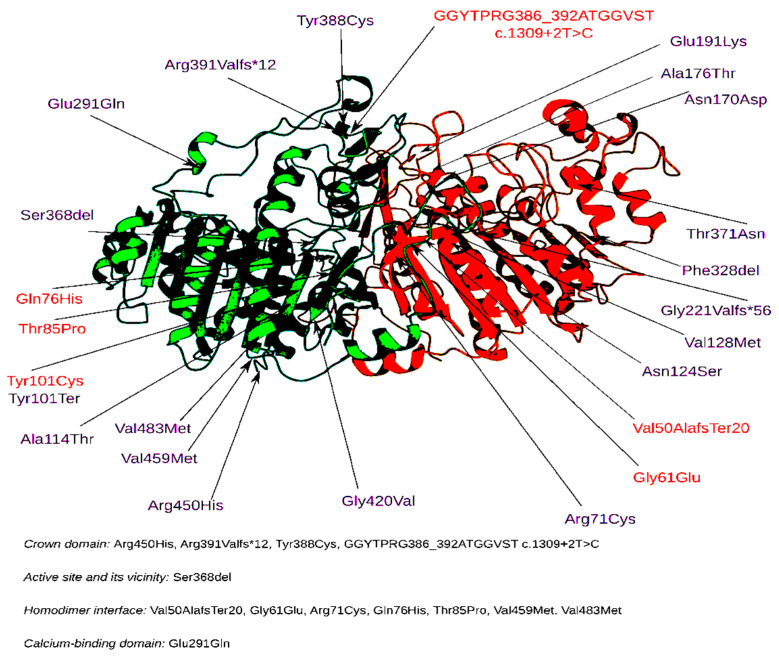
Localization of amino acids in TNSALP protein domains in Russian patients (compound heterozygotes only). The altered amino acids identified in the Russian population are highlighted in red.

**Figure 8 ijms-23-12976-f008:**
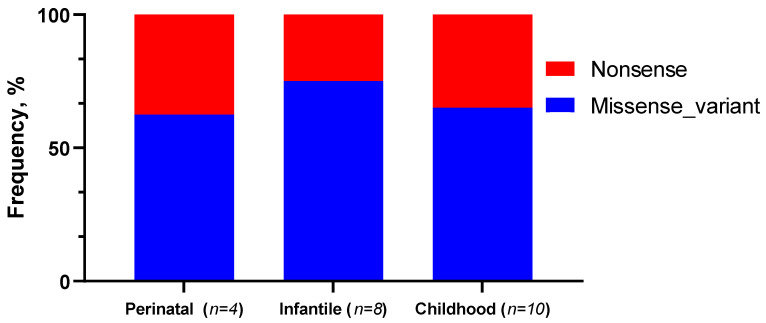
Frequencies of genetic variants in the *ALPL* gene depending on the HPP forms in the patients.

**Figure 9 ijms-23-12976-f009:**
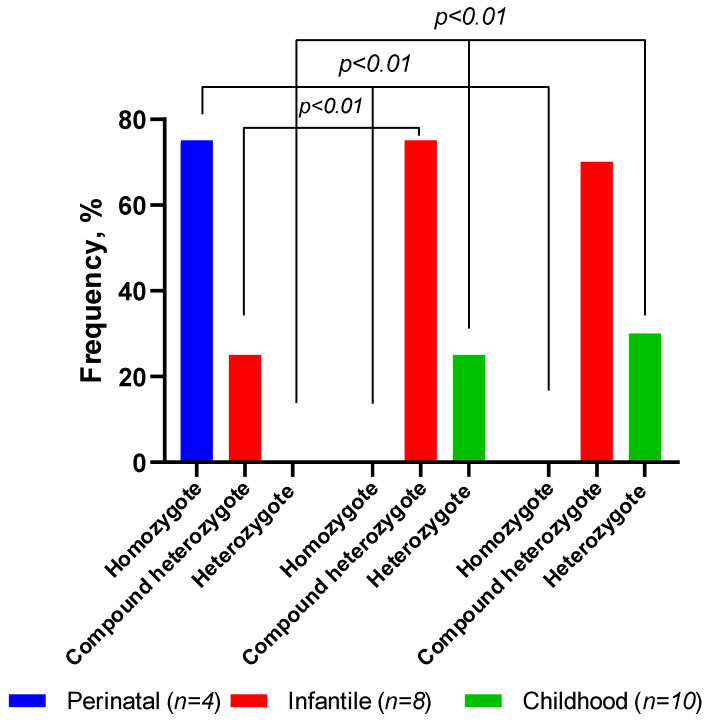
Frequency of the homozygous, compound-heterozygote, and heterozygote genotypes of the *ALPL* gene in patients, depending on the form of HPP.

**Figure 10 ijms-23-12976-f010:**
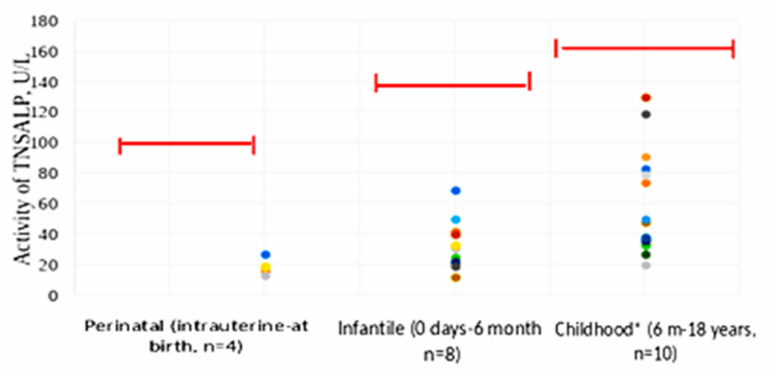
TNSALP activity, depending on the form of HPP in patients (*n* = 22). Each colored dot in the figure explain one patient. Note: each point is one patient. Red and black indicate the lower limit of normal TNSALP at this age. * Most patients with the childhood form were diagnosed at the age of 2–9 years. The middle limit of normal was taken from the study [23].

**Figure 11 ijms-23-12976-f011:**
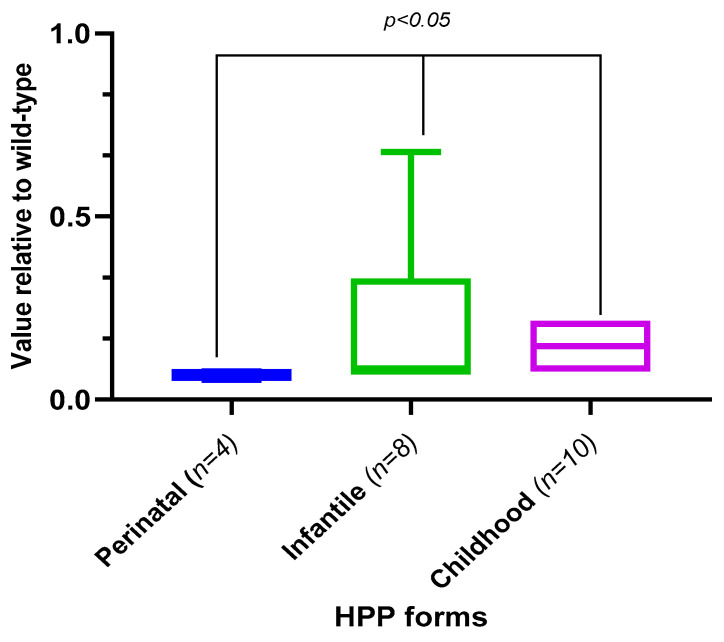
Residual activity of TNSALP variants relative to their wild-type, according to HPP subtype [21].

**Figure 12 ijms-23-12976-f012:**
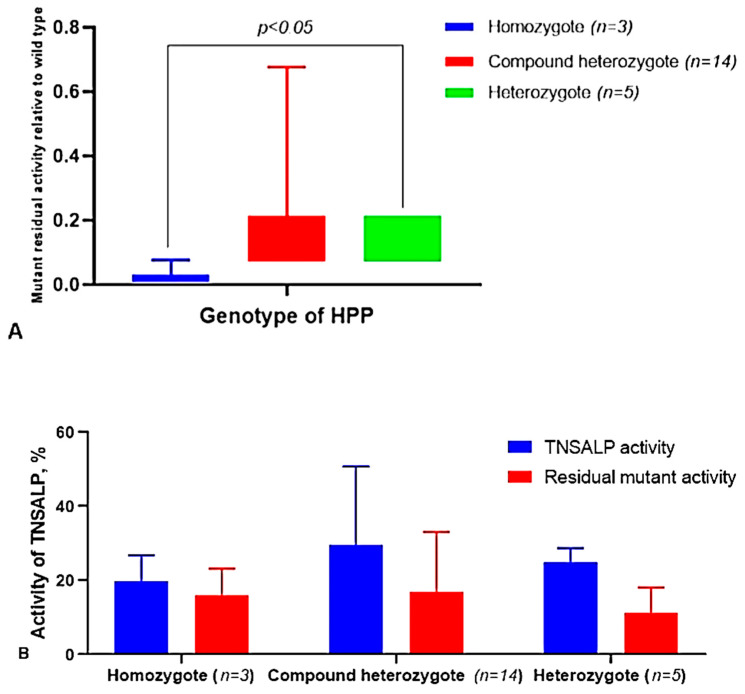
The residual mutant activity of the different variants of the *ALPL* gene in relation to wild-type activity deepened according to genotype. (**A**) Comparison of the real TNSALP activity in the serum with the theoretically calculated values, depending on the genotypes of the patients with HPP ((**B**), see the Material and Methods section).

**Figure 13 ijms-23-12976-f013:**
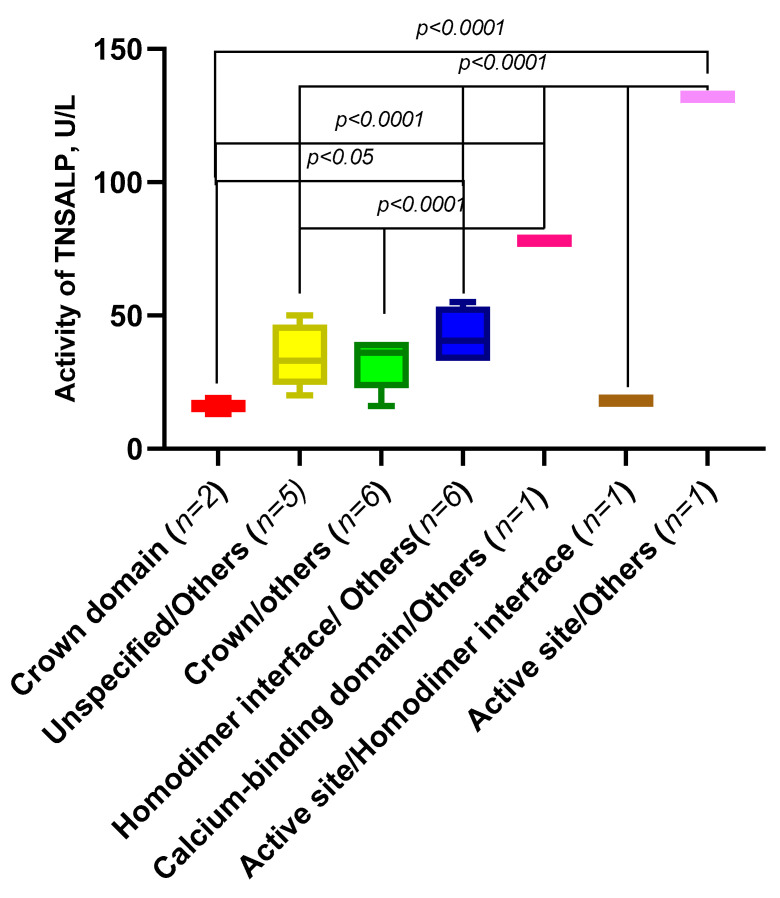
Activity of *ALPL* gene variants, depending on the protein domain location.

**Table 1 ijms-23-12976-t001:** Clinical forms of HPP in patients with two variants of the *ALPL* gene (*n* = 22).

HPP Onset	Number of Patients	Age at the Time of Diagnosis, Years	TNSALP Level in Serum, U/L
Prenatal	4	up to 1 year	18.7 ± 6.0
Infantile	8	2.0 ± 1.2 (1–4)	31.6 ± 10.8
Childhood	10	5.3 ± 5.0 (1–18)	40.1 ± 16.5

**Table 2 ijms-23-12976-t002:** Genetic variants of the *ALPL* gene in 27 compound heterozygotes and homozygotes HPP patients.

Allele 1	Allele 2	Amino Acid Substitution	Variant Type	Number of Patients	Clinical Varinat Classification
c.182G > A	c.571G > A	Gly61Glu/Glu191Lys	Missense/Missense	1	Pathogenic/Pathogenic
c.211C > T	c.1101_1103del	Arg71Cys/Ser368del	Missense/In frame	1	Pathogenic/Pathogenic
c.228G > T	c.571G > A	Gln76His/Glu191Lys	Missense/Missense	1	Pathogenic/Pathogenic
c.253A > C	c.571G > A	Thr85Pro/Glu191Lys	Missense/Missense	1	VUS/Pathogenic
c.302A > G	c.571G > A	Tyr101Cys/Glu191Lys	Missense/Missense	1	VUS/Pathogenic
c.303C > A	c.314C > T	Tyr101Ter/Ala105Val	Nonsense/Missense	1	Pathogenic/VUS
c.303C > A	c.571G > A	Tyr101Ter/Glu191Lys	Nonsense/Missense	1	Pathogenic/Pathogenic
c.340G > A	c.571G > A	Ala114Thr/Glu191Lys	Missense/Missense	1	Pathogenic/Pathogenic
c.382G > A	c.871G > C	Val128Met/Glu291Gln	Missense/Missense	1	Pathogenic/VUS
c.508A > G	c.508A > G	Asn170Asp	Missense	1	Pathogenic
c.526G > A	c.1375G > A	Ala176Thr/Val459Met	Missense/Missense	1	Pathogenic/Pathogenic
c.571G > A	c.144_148dup	Glu191Lys/	Missense/Frameshift	1	Pathogenic/VUS
		Val50AlafsTer20			
c.571G > A	c.662delG	Glu191Lys/	Missense//Frameshift	2	Pathogenic/Pathogenic
		Gly221Valfs*56			
c.571G > A	c.1259G > T	Glu191Lys/Gly420Val	Missense/Missence	1	Pathogenic/Pathogenic
c.571G > A	c.1157_1175inv	Glu191Lys/GGYTPRG386_392ATGGVST	Missense/Frameshift	1	Pathogenic//Pathogenic
c.571G > A	c.1309 + 2T > C	Glu191Lys/	Missense/Splicing	2	Pathogenic/Pathogenic
c.984_986delCT	c.371A > G	Phe328del/Asn124Ser	In frame/Missense	1	Pathogenic/VUS
c.1112C > A	c.1447G > A	Thr371Asn/Val483Met	Missense/Missense	2	Pathogenic/VUS
c.1163A > G	c.1163A > G	Tyr388Cys	Missense	1	VUS
c.1171del	c.1171del	Arg391ValfsTer12	Frameshift	1	Pathogenic
c.1171delC	c.302A > G	Arg391Valfs*12/	Fameshift/Missense	1	Pathogenic/VUS
		Tyr101Cys			
c.1171delC	c.571G > A	Arg391Valfs*12/Glu191Lys	Fameshift/Missense	1	Pathogenic/Pathogenic
c.1349G > A	c.1349G > A	Arg450His	Missense	1	Pathogenic
c.1364G > A	c.571G > A	Glu455Asp/Glu191Lys	Missense/Missense	1	Pathogenic/Pathogenic

**Table 3 ijms-23-12976-t003:** Genetic variants of the *ALPL* gene in 225 heterozygote HPP patients.

Variant	Amino Acid Substitution	Variant Type	Allelic	Clinical Variants
			Frequency, %	Interpretation
c.61G > A	Glu21Lys	Missense	0.22 (1/450)	Pathogenic
c.98C > T	Ala33Val	Missense	0.22 (1/450)	Pathogenic
c.119C > T	Ala40Val	Missense	0.22 (1/450)	Pathogenic
c.144_148dup	Val50Alafs*20	Frameshift	0.22 (1/450)	Pathogenic
c.182G > A	Gly61Glu	Missense	0.22 (1/450)	VUS
c.188G > T	Gly63Val	Missense	0.44 (2/450)	Pathogenic
c.202_204delACG	Thr68del	In frame	0.22 (1/450)	VUS
c.203C > T	Thr68Met	Missense	0.44 (2/450)	Pathogenic
c.205G > A	Ala69Thr	Missense	1.11 (5/450)	VUS
c. 211C > T	Arg71Cys	Missense	0.67 (3/450)	Pathogenic
c.211C > A	p.Arg71Ser	Missense	0.22 (1/450)	Pathogenic
c.212G > A	p.Arg71His	Missense	0.22 (1/450)	Pathogenic
c.214A > G	Ile72Val	Missense	0.22 (1/450)	Likely Pathogenic
c.216C > A	p.(=)	Synonymous	0.22 (1/450)	VUS
c.250G > A	Glu84Lys	Missense	0.22 (1/450)	Pathogenic
c.253A > C	Thr85Pro	Missense	0.22 (1/450)	VUS
c.297 + 1G > A	-	Splicing	0.44 (2/450)	Pathogenic
c.298-2A > G	-	Splicing	0.22 (1/450)	Pathogenic
c.302A > G	Tyr101Cys	Missense	0.22 (1/450)	VUS
c.303C > A	Tyr101Ter	Nonsense	1.11 (5/450)	Pathogenic
c.305A > C	Asn102Thr	Missense	0.44 (2/450)	VUS
c.314C > T	Ala105Val	Missense	0.22 (1/450)	VUS
c.331_332insCCGGCA	p.T113_A114insGT	In frame	0.22 (1/450)	VUS
c.368C > A	Ala123Asp	Missense	0.44 (2/450)	Pathogenic
c.371A > G	Asn124Ser	Missense	0.22 (1/450)	VUS
c.389del	Val130GlufsTer35	Frameshift	0.22 (1/450)	Pathogenic
c.407G > A	Arg136His	Missense	0.22 (1/450)	Pathogenic
c.436G > A	Glu146Lys	Missense	0.67 (3/450)	VUS
c.455G > A	Arg152His	Missense	3.33 (15/450)	VUS
c.500C > T	Thr167Met	Missense	0.22 (1/450)	Pathogenic
c.508A > G	Asn170Asp	Missense	0.44 (2/450)	Pathogenic
c.571G > A	Glu191Lys	Missense	12.7 (57/450)	Pathogenic
c.595C > T	Gly199*	Nonsense	0.44 (2/450)	Pathogenic
c.648 + 1G > A		Splicing	0.22 (1/450)	Pathogenic
c.659G > T	Gly220Ala	Missense	0.67 (3/450)	Pathogenic
c.662_663insG	Gly222Trpfs	Frameshift	0.44 (2/450)	Pathogenic
c.815G > T	Arg272Leu	Missense	0.22 (1/450)	Pathogenic
c.818C > T	Thr273Met	Missense	0.88 (4/450)	VUS
c.889T > A	Tyr297Asn	Missense	0.44 (2/450)	VUS
c.902delG	Asn302Thrfs	Frameshift	0.67 (3/450)	Pathogenic
c.928_929delTC	Ser310Argfs	Frameshift	0.44 (2/450)	Pathogenic
c.968A > T	Asn323Ile	Missense	0.22 (1/450)	VUS
c.979T > C	Phe327Leu	Missense	0.22 (1/450)	Pathogenic
c.984_986delCTT	Phe328del	In frame	0.22 (1/450)	Pathogenic
c.997 + 2T > A	-	Splicing	0.44 (2/450)	Pathogenic
c.998-1G > A	-	Splicing	0.67 (3/450)	Pathogenic
c.1001G > A	Gly334Asp	Missense	0.44 (1/225)	Pathogenic
c.1066G > A	Asp356Asn	Missense	0.22 (1/450)	VUS
c.1072G > A	Ala358Thr	Missense	0.22 (1/450)	VUS
c.1101_1103del	Ser368del	In frame	0.22 (1/450)	Pathogenic
c.1130C > T	Ala377Val	Missense	0.22 (1/450)	Pathogenic
c.1163A > G	Tyr388Cys	Missense	0.22 (1/450)	VUS
c. 1171del (16)	Arg391Valfs*12	Frameshift	3.11 (14/450)	Pathogenic
c. 1190-2_1190-1delinsCT	-	Splicing	0.44 (2/450)	Pathogenic
c.1216G > A	Asp406Asn	Missense	0.22 (1/450)	VUS
c.1247G > T	Gly416Val	Missense	0.22 (1/450)	VUS
c.1259G > T	Gly420Val	Missense	0.44 (2/450)	VUS
c.1276G > A	p.Gly426Ser	Missense	0.44 (2/450)	Pathogenic
c.1309 + 2T > C	-	Splicing	0.44 (2/450)	Pathogenic
c.1310C > T	Ala437Val	Missense	0.22 (1/450)	VUS
c.1328C > T	Ala443Val	Missense	0.22 (1/450)	Pathogenic
c.1349G > A	Arg450His	Missense	0.67 (3/450)	Pathogenic
c.1364G > A	Glu455Asp	Missense	0.44 (2/450)	Pathogenic
c.1447G > A	Val483Met	Missense	1.11 (5/450)	VUS
c.1483G > A	Gly495Ser	Missense	0.22 (1/450)	VUS
c.1489T > C	Cys497Arg	Missense	0.22 (1/450)	VUS

**Table 4 ijms-23-12976-t004:** Correlation of clinical symptoms with variants in the *ALPL* gene in Russian patients with perinatal, infantile, and childhood forms of HPP.

Clinical Symptoms	Patients, Forms of HPP																					
	Perinatal				Infantile								Childhood									
Loss of teeth																						
Skeletal deformities																						
Hypotension																						
Delayed motor development																						
Malnutrition (probe, regurgitation)																						
Funnel chest																						
Craniosynostosis																						
Gait disturbance																						
Fast fatiguability																						
Muscular dystonia																						
Pain in muscles and joints																						
Joint hypermobility																						
Valgus deformity of the knee joints																						
Impaired mental function																						
Hypomineralization of the bones																						
Skull deformities																						
short stature																						
fractures																						
Hypomineralization of bone tissue																						
Respiratory failure																						
limb shortening																						
Hypercalcemia																						
Planovalgus foot deformity																						
Osteoporosis																						
Protein-energy malnutrition																						
convulsions																						
Nephrocalcinosis/kidney damage																						
Hypercalciuria																						
Hyperphosphatemia																						
Tubulopathy																						
Hydrocephalus																						
Signs of bone defects																						
Poorly healed fractures of the tibia																						
Anomaly Arnold Chiari II,																						
Multiple organ failure																						
Frequent colds, bronchitis																						
Irritability																						
Tremor																						
Myopia																						
Complex astigmatism																						
Genotype	c.1163A>G/c.1163A>G	c.1171delC/c.1171delC	c.1171delC/c.302A>G	c.508A>G/c.508A>G	c.571G>A/c.1259G>T	c.303C>A/c.314C>T	c.253A>C/c.571G>A	c.571G>A/c.1157_1175inv	c.984_986delCT/c.371A>G	c.526G>A/c.1375G>A	c.211C>T/c.1101_1103del	c.1364G>A/c.571G>A	c.340G>A/c.571G>A	c.303C>A/c.571G>A	c.571G>A/c.144_148dup	c.571G>A/c.1309+2T>C	c.571G>A/c.1309+2T>C	c.382G>A/c.871G>C	c.571G>A/c.662delG	c.571G>A/c.662delG	c.302A>G/c.571G>A	c.1171delC/c.571G>A


Note: Color 

 indicates homozygotes, 

—compound heterozygotes.

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
