# Peer review of "Clinical and Genetic Characteristics of Pediatric Patients with Hypophosphatasia in the Russian Population"

_ijms, 2022, doi:10.3390/ijms232112976_

Round 1

Reviewer 1 Report

In this article, Glotov et al explore the association between the clinical and genetic characteristics of Russian patients with hypophosphatasia, the disease caused by mutations in tissue nonspecific alkaline phosphatase (TNAP).

This is an impressive work on 1612 patients with reduced blood TNAP activity, providing interesting results. I only have a few questions:

-          In the introduction, authors write that TNAP dephosphorylates phosphoethanolamine (PEA). If it is true that PEA levels are increased in hypophosphatasia, I think it is still unknown whether TNAP directly dephosphorylates PEA, or whether decreased PEA levels result from indirect effects, in particular associated with pyridoxal-dependent enzymes. This might be reformulated.

-          Epileptic seizures is not reported as a frequent symptom in the Russian population, appearing as rare in patients in Table 3 for instance. Is there a particular reason? Is it different from other populations? If some patients experienced seizures, how many patients were treated with vitamin B6 to prevent seizures?

-          Table 3 reports twice Hypomineralization of bone tissue, with different cases, I do not understand why. Maybe it is because the first line refers to the skull?

-          In figure 8B, authors should better explain in the legend or in the text, where TNAP activity was measured

Minor points:

-          ALPL gene is sometimes but not always in italics.  

Author Response

Dear Reviewer,

We thank you for your carefully attention to our article. You will find answers to your questions and comments in attacted file, as well as in the manuscript, where the corrections are highlighted in color. 

Reviewer 2 Report

Glotov et al, reported a genetic and clinical study of HPP pediatric patients

The work adds novel insights on the topic. However, the whole text needs to be carefully revised by a native English speaker.

Major revisions.

For this Reviewer, the major concern lies in the Conclusions, where the Authors state: “it is also possible that the different frequency of symptoms is due to the different criteria for confirming this symptom in Russia and abroad”.

The Authors need to clarify the recruitment exclusion/inclusion criteria and how a clear-cut diagnosis of HPP was reached.

In fact, it is not clear the sample size of the cohort under study. The Authors initially recruited 1612 patients with reduced alkaline phosphatase and it seems they screened them all for ALPL gene mutations.

In the end, they found 225 heterozygous and 25 homozygous/compound heterozygous patients, for a total of 252 mutated ones. If this is correct, the mutation rate would be around the 15% [(225+27)/1612], an unusual low frequency compared to other studies. Thus, it would be of help, if the Authors clarify if (and how) they reduced the initial sample size (of 1612 pts) by a further deep clinical analysis, to a well clinically ascertained HPP cohort.

About the list of the variants, the Authors declared they followed the ACMG criteria to define the benignity/pathogenicity level. However, for most of the variants, the interpretation was not correct.

Minor revisions

Some words or even some whole sentences are in bold. Why?

Abstract.

Line 28: encodes instead of “encoding”: please correct.

Line 29: HPP “is” characterized.

Line 38: it would preferable to avoid abbreviations in the abstract: please clarify the term DBS.

Line 39 and 40: this sentence is part of the methods: please delete or move it in the corresponding section.

Line 43: “two pathogenic variants…”: for many homozygous/compound heterozygous patients, the variants are not both pathogenic: please correct.

Results

This Reviewer suggests adding a separate table as a Suppl Mat, including for all the variants, the corresponding interpretation criteria (PM, PP, BS and so on) used for classification.

Lines 132-139 and 146-152: in light of the correction of the interpretation as suggested above, these parts need to be revised accordingly.

Line 178: “there were also 6 mutations located…”: please define these variants.

Line 195: please explicit the term “PM”.

Lines 205-217: it is not clear to this Reviewer the aim of these statistical analysis. Instead, the Authors want to demonstrate that perinatal forms were mostly characterized by full ALPL insufficiency, while milder forms by haploinsufficiency. However, this result is not novel and is pretty expected.

And, the same expected conclusion can be applied on both the Figures 6 and 7, where the Authors show that the residual enzymatic activity is lower in severe forms compared to the milder ones.

Line 224: “Red black” is not clear, please clarify.

Lines 225-226: the sentence is blurry, please re-write.

Legends

Table 1: the table does not include only the 27 compound heterozygotes, but also the homozygous. Please correct. “In bold are options not available…”is not clear. Please clarify.

Figure 1: the figure does not show the “localization” of the variants. Please correct.

Figure 2: in light of the correction of the interpretation as suggested above, the whole figure needs to be modified.

Figure 6: it is not clear and it is seems not to be described into the text, how the Authors calculated theoretically the residual ALPL enzymatic value. Please clarify.

Table 4: the variants identified in these patients are not necessarily both “pathogenic”. Please correct the table accordingly to the novel interpretation suggested above.

Discussion

Lines 253-267: this first part needs to be shortened: instead, it is a mere comparison of the clinical symptoms found in HPP patients belonging to different countries.

Line 280: “two pathogenic variants”: please see comment above.

Line 304: the c.787T>C, p.(Tyr263His) variant is a classic known SNP, and, probably, it should not be considered.

Line 312: following the line of reasoning, the term “pathogenicity” should be substitute with “benignity”.

Line 317: what does “most genetic variants” mean? Please clarify.

Line 345: the term “fixation” is not clear. Please clarify.

Material and Methods

Lines 366-379: these sections should be part of the Results.

Line 387: please describe in detail the assay used for the TNSALP activity in the serum.

Line 388: “two pathogenic variants”: please see comment above.

Line 405: “specify which was, usually Tukey” is not clear: please correct.

Conclusions

Lines 418-424. This section should be deleted, since it is a mere description of the results obtained.

Lines 426-429: the sentence needs to be re-write.

Author Response

Dear Reviewer,

We thank you for your carefully attention for our article. You will find answers to your comments and questions in attached file, as wel as in the manuscript, where corrections are highlighted in color. 
